# Evidencing leprosy neuronal inflammation by 18-Fluoro-deoxy-glucose

**Patricia Sola Penna**[1]*, **Sergio Augusto Lopes De Souza**[2], **Paulo Gustavo Limeira Nobre De Lacerda**[2], **Izabela Jardim Rodrigues Pitta**[3], **Clarissa Neves Spitz**[1,3,4], **Anna Maria Sales**[3], **Flavio Alves Lara**[5], **Ana Caroline Siquara De Souza**[1,3], **Euzenir Nunes Sarno**[3], **Roberta Olmo Pinheiro**[3,6,7], **Marcia Rodrigues Jardim**[1,3,4,6,7]

1 Universidade Federal do Estado do Rio de Janeiro, PPGNeuro, Rio de Janeiro, Brazil, 2 Universidade Federal do Rio de Janeiro, Departamento de Radiologia, Serviço de Medicina Nuclear, Rio de Janeiro, Brazil, 3 Instituto Oswaldo Cruz, Laboratório de Hanseníase (LAHAN), Rio de Janeiro, Brazil, 4 Universidade do Estado do Rio de Janeiro, Departamento de Neurologia, Rio de Janeiro, Brazil, 5 Instituto Oswaldo Cruz, Laboratório de Microbiologia Celular (LAMICEL), Rio de Janeiro, Brazil, 6 Rio de Janeiro Research Network on Neuroinflammation, Oswaldo Cruz Institute, Oswaldo Cruz Foundation, Rio de Janeiro, Brazil, 7 National Institute of Science and Technology on Neuroimmunomodulation, Oswaldo Cruz Institute, Oswaldo Cruz Foundation, Rio de Janeiro, Brazil

* patsolapenna@gmail.com

**Data Availability Statement:** Patient data are available in the Oswaldo Cruz Foundation Leprosy Laboratory database. Adress: Avenida Brasil, 4025-4011, Manguinhos, Rio de Janeiro - RJ Brazil.

## Abstract

### Background

Leprosy is caused by multiple interactions between *Mycobacterium leprae (M. leprae)* and the host's peripheral nerve cells. *M. leprae* primarily invades Schwann cells, causing nerve damage and consequent development of disabilities. Despite its long history, the pathophysiological mechanisms of nerve damage in the lepromatous pole of leprosy remain poorly understood. This study used the findings of 18F-FDG PET/CT on the peripheral nerves of eight lepromatous patients to evaluate the degree of glucose uptake by peripheral nerves and compared them with clinical, electrophysiological, and histopathological evaluations.

### Methods

Eight patients with lepromatous leprosy were included in this study. Six patients were evaluated up to three months after leprosy diagnosis using neurological examination, nerve conduction study, 18F-FDG PET/CT, and nerve biopsy. Two others were evaluated during an episode of acute neuritis, with clinical, neurophysiological, and PET-CT examinations to compare the images with the first six.

### Results

Initially, six patients already had signs of peripheral nerve injury, regardless of symptoms; however, they did not present with signs of neuritis, and there was little or no uptake of 18F-FDG in the clinically and electrophysiologically affected nerves. Two patients with signs of acute neuritis had 18F-FDG uptake in the affected nerves.

phone +55 21 25621588. Website: https://www.
ioc.fiocruz.br/en/referencia/hanseniase. E-mail: asa.
lahan@ioc.fiocruz.br PET/CT images are available
in the database of the Nuclear Medicine Lab of the
Federal University of Rio de Janeiro. at Hospital
Universitário Clementino Fraga Filho Phone +55 21
39382362 Website: https://www.hucff.ufrj.br/.

**Funding:** The author(s) received no specific
funding for this work.

**Competing interests:** The authors have declared
that no competing interests exist.

## Conclusions

18F-FDG uptake correlates with clinical neuritis in lepromatous leprosy patients but not in
silent neuritis patients. 18F-FDG PET-CT could be a useful tool to confirm neuritis, espe-
cially in cases that are difficult to diagnose, such as for the differential diagnosis between a
new episode of neuritis and chronic neuropathy.

## Author summary

The lepromatous pole (LL) is the prototype of leprosy patients, contributing to its stigma.
Nerve function impairment in LL patients may be clinically asymptomatic for an extended
period, and despite its long history, lepromatous leprosy nerve damage pathophysiological
mechanisms remain poorly understood. Tissues involved in infectious and inflammatory
diseases are hypermetabolic and have increased glucose uptake by 18F-FDG. Our question
was about how the uptake of this radiopharmaceutical would be in LL patients' silent
neuritis.

Our data demonstrated, for the first time, systemic peripheral nerve 18F-FDG incorpo-
ration using PET/CT whole-body images of a group of lepromatous patients with silent
neuritis and acute neuritis. 18F-FDG PET-CT could be a useful tool to confirm neuritis,
especially in cases that are difficult to diagnose, such as for the differential diagnosis
between a new episode of neuritis and chronic neuropathy.

## Introduction

Lepromatous leprosy (LL) is characterized by scattered skin lesions, which are more frequent
in the cooler areas of the body with high bacterial load due to the host's modest or absent cell-
mediated immunity against the infectious agent *Mycobacterium leprae (M. leprae)* [1]. Periph-
eral neuropathy in patients with LL is characterized by the uninhibited multiplication of bacte-
ria, predominantly in Schwann cells (SC), due to a series of immune and metabolic
subversions of host cells imposed by the pathogen [2,3], which will, in the last instance, cause
leprosy neuropathy. Nerve function impairment in patients with LL may be clinically asymp-
tomatic for an extended period, even if microscopically involved, and may present clinical and
electrophysiological evidence of extensive peripheral nerve damage from the onset of infection
[4].

This condition of asymptomatic or oligosymptomatic sensory or sensory-motor nerve dys-
function that evolves indolently and is common in patients with LL is called silent neuritis
[5,6]. Some authors reported that, in terms of nerve conduction studies, LL patients' overall
condition worsened, and abnormalities persisted despite improvements in skin lesions follow-
ing multidrug therapy (MDT), even in patients without evident neuritis [4,7,8] or in those
receiving corticosteroids [8]. Experimental *in vivo* and *in vitro* studies have shown that *M.
leprae* invades SC early and can reprogram host gene expression [9].

Medeiros *et al.* (2016) evaluated *in vitro* and *in vivo* models of Schwann cells infected with
*M. leprae*. They concluded that the bacillus alters the metabolism of Schwann cells, reducing
mitochondrial activity and increasing glucose uptake by up to 50% [3]. Recently, this anaerobic
shift of infected Schwann cells was found to be related to lipid accumulation, *M. leprae*'s eva-
sion of the innate immune response, and axonal loss of function [10].

18-Fluoro-deoxy-glucose (18F-FDG) is a well-known radiopharmaceutical used in Positron Emission Tomography (PET/CT) that mimics glucose in the cell's entry mechanisms, using glucose transposing proteins (Glut), just like non-radioactive glucose. Tissues involved in infectious and inflammatory diseases, with high expression of neutrophils and activated macrophages, demonstrate increased glucose uptake and 18F-FDG. Roy et al. (2018) and Shao et al. (2020) described case reports of acute neuritis in leprosy patients detected using 18F-FDG PET/CT [11,12].

Herein, we describe systemic peripheral nerve 18F-FDG incorporation using PET/CT whole-body images of eight lepromatous patients, correlating the degree of peripheral nerve glucose uptake with clinical, electrophysiological, and histopathological data.

This research study aimed to evaluate the extent of infection/inflammation and infer possible injury mechanisms in the peripheral nerves of LL patients with silent neuritis using an imaging method that mimics the entry of glucose into inflamed tissues.

## Methods

### Ethics statements

The research was conducted in compliance with the international compilation of human research standards, which was previously approved by the Ethics Committee of the Oswaldo Cruz Foundation (Approval number:3.152.162). All the patients provided written informed consent.

### Patient selection

This is a case series of eight patients evaluated by the neurology team from the Souza Araújo Outpatient Clinic of the Leprosy Laboratory at the Oswaldo Cruz Foundation (Fiocruz) in Rio de Janeiro, Brazil.

From February 2019 to May 2022, 42 patients with the LL leprosy were admitted to the Souza Araújo outpatient clinic. Among these, 34 were excluded from the study: nine due to reactions related to leprosy; 10 had comorbidities associated with peripheral neuropathy, such as diabetes mellitus (eight patients), chronic alcoholism (one patient), and hypothyroidism (one patient); 12 patients did not accept to participate in the study; one was less than 18 years of age; one was pregnant; one was a case of recurrent leprosy.

Eight patients participated in the study; six were evaluated at the beginning of treatment and up to three months after diagnosis. The remaining two were assessed during clinical follow-up for episodes of acute neuritis.

### Clinical and electrophysiological evaluation

All patients were diagnosed with the LL form of leprosy according to the Ridley and Jopling Classification based on the results of their slit-skin smears and skin histopathology. Patients underwent clinical examinations for leprosy diagnosis according to the protocol of the Leprosy Outpatient Unit of the Oswaldo Cruz Institute [13].

Six patients (cases 1–6) were in the initial phase of treatment. Patient 7 had already completed the MDT; he was submitted to neurological and nerve conduction evaluation at diagnosis but had been presenting Erythema Nodosum Leprosum (ENL) reactions since the beginning of the treatment, so it was initially excluded; furthermore, he developed acute neuritis two years after admission to the Souza Araújo Outpatient Clinic and was included for 18F-FDG PET/CT comparison. Patient 8 underwent treatment for six months and underwent a neurological examination one month following diagnosis; however, he did not accept

participating in the study then. Six months later, when the patient experienced an acute neuritis episode, he was included in the 18F-FDG PET/CT evaluation.

Neurological examinations and nerve conduction studies were performed according to the protocol of the Leprosy Outpatient Unit of the Oswaldo Cruz Institute, published elsewhere [14]. The disability grade was recorded using the standard WHO grading criteria [15]. Nerve function impairment was defined as clinically detectable impairment of motor, sensory, and autonomic functions [16].

To evaluate the extent of nerve involvement, neuropathy was classified according to the number of impaired nerves and distribution of impairment in nerve conduction. Polyneuropathy was defined as the presence of diffused symmetrical peripheral nerve lesions. In addition, patients were diagnosed with mononeuropathy when a single nerve was affected and with multiple mononeuropathies when there was focal involvement of two or more nerves [17].

Based on the compound muscle and sensory nerve action potentials, the nerve segment lesion pathophysiology categories were defined by combining nerve conduction study parameters. In short, an axonal lesion was defined as either an isolated reduction in amplitude $\geq$ 30% of the reference values or a combined change in amplitude reduction of less than 30% associated with a reduction in conduction velocity of up to 60–75% of the reference values. Demyelination was verified as a $\geq$ 20% increase in latency, a > 35% reduction in conduction velocity, or a combined decrease in amplitude of up to 20% together with a 15–20% increased latency. Demyelinating lesions with axonal degeneration were determined by axonal and demyelinating lesions within the same nerve. A lesion was considered "no conduction" when action potentials could not be recorded [4].

## PET/CT imaging evaluation

PET/CT was performed at the Nuclear Medicine Service of Clementino Fraga Filho University Hospital of the Federal University of Rio de Janeiro. All patients underwent examination following a 24 h low carb diet. After 60 min of intravenous administration of 0.14 mCi/kg of 18 F-fluorodeoxyglucose, the whole-body PET/CT examination was performed, and sequential images of computed tomography (16-channel CT) without contrast and PET were obtained, with the area of interest being a strip that extended from the cranial cap to the root of the thighs. In addition, the second group of images was acquired from the lower limbs. All images were reviewed in the transaxial, coronal, and sagittal planes and semi-quantitative analysis was performed using the target maximum standardized capture value (t-SUV max), considering the patient's body mass as an index, and compared with the background SUVmax (bg-SUVmax).

## Histopathological evaluation

The biopsied nerve samples were analyzed at the Leprosy Laboratory of Oswaldo Cruz Institute in Rio de Janeiro, Brazil. The sensory nerve was biopsied according to clinical and electrophysiological findings in six patients evaluated at the beginning of treatment (cases 1–6), not necessarily the same ones that showed 18F-FDG uptake. Furthermore, the following nerves were biopsied: dorsal cutaneous ulnar nerve on the dorsum of the hand (n = 4) and sural nerve at the ankle level (n = 2). Nerve samples were analyzed according to standard methods [18].

## Results

Table 1 summarizes the demographic and clinical characteristics of the eight patients. Seven patients were male, and one was female; the ages ranged from 21 to 70 years (mean age, 42.6 years). The bacilloscopic index (BI) ranged from 3.5–5.75 (mean 5.08).

**Table 1. Clinical characteristics of recruited patients.**

| Patient | BI | Symp on (y) | Pain | Par | Thick | Tact | Ther | Painf | Motor | Neuritis |
|---------|------|-------------|------|-----|-------|------|------|-------|-------|----------|
| 1 | 5 | No | No | No | No | No | No | No | No | No |
| 2 | 5.5 | 2 | Yes | Yes | Yes | Yes | Yes | Yes | Yes | No |
| 3 | 5.25 | No | No | No | Yes | No | Yes | Yes | No | No |
| 4 | 5.75 | 3 | No | Yes | No | Yes | Yes | Yes | No | No |
| 5 | 3.5 | 2 | Yes | Yes | No | Yes | Yes | Yes | Yes | No |
| 6 | 5.5 | No info | Yes | Yes | Yes | Yes | Yes | Yes | Yes | No |
| 7 | 5.5 | No info | Yes | Yes | Yes | Yes | Yes | Yes | Yes | Yes |
| 8 | 5 | 0 | Yes | No | Yes | Yes | Yes | Yes | No | Yes |

Abbreviations: BI, baciloscopic Index; Symp on (y), referred neural symptoms in years; No info, not informed; Par, Paresthesia; Thick—Thickening Tact, Tactile impairment; Ther, Thermal impairment; Painf, painful impairment; Motor, Motor impairment.

On neurological examination, six out of eight (75%) patients had symptoms of peripheral nerve damage (pain and paresthesia), two (25%) reported the onset of nerve symptoms before the dermatological injury (cases 4 and 7), two (25%) reported nerve symptoms and dermatological injury at the same time (cases 5 and 6), and two (25%) reported the onset of neurological symptoms after the dermatological lesions (cases 2 and 8). Patients 1 and 3 were asymptomatic at the time of examination.

## Six patients evaluated at the beginning of MDT presented signs of peripheral nerve damage, regardless of symptoms

Of the six patients evaluated at the beginning of treatment (cases 1–6), four (66.6%) had symptoms related to peripheral nerve damage; paresthesia was the most common (66.6%). Neurological examination showed signs of peripheral nerve damage in five patients (83.3%). Thickening was observed in three of six patients (50%), sensory impairment was observed in five patients (83.3%), and motor impairment was observed in three patients (50%) (Table 1).

Table 2 summarizes the correlation between the eight patients' clinical, electrophysiological, and PET/CT uptake findings.

Patients 1, 3, and 5 showed no uptake of 18F-FDG in any peripheral nerve but had diffuse electrophysiological alterations in a pattern of multiple mononeuropathies. The neurological examination was normal in patient 1; however, there were signs of peripheral nerve involvement in patients 3 and 5 (Fig 1).

Patients 2, 4, and 6 had 18F-FDG uptake in one to two peripheral nerves each, while nerve conduction studies and neurological examination findings demonstrated a more diffuse pattern of alteration, such as polyneuropathy (Fig 2).

In patients 7 and 8, PET/CT was performed during an episode of acute neuritis. 18F-FDG uptake has been observed in the same symptomatic nerves and other clinically asymptomatic nerves, with nerve damage evidence in nerve conduction studies.

One year after MDT, Patient 7 developed worsening pain, paresthesia, and decreased loss of muscle strength in the right foot. All evaluated sensory and motor nerves were unexcitable to stimuli in nerve conduction studies. 18F-FDG PET/CT revealed hypermetabolism of radiolabeled glucose in the topography of the left sural and right deep peroneal nerves.

Patient 8 developed severe pain in the left ulnar path six months after the leprosy diagnosis. We observed bilateral lesions in the sensory and motor ulnar nerves clinically and through nerve conduction studies. In addition, the sensory nerves of the lower limbs were not excitable

**Table 2. Correlation between clinical findings, NCS findings and PET uptake.**

| | Clinical | NCS | t-SUV max/bg-SUV max | Clinical | NCS | t-SUV max/bg-SUV max | Clinical | NCS | t-SUV max/bg-SUV max |
|---|---|---|---|---|---|---|---|---|---|
| **Patient** | | | **1** | | | **2** | | | **3** |
| **Nerve** | | | | | | | | | |
| R Median | normal | axonal | no activ | abnormal | 0 | no activ | normal | axonal | no activ |
| R Radial | normal | normal | no activ | abnormal | 0 | no activ | normal | axonal | no activ |
| R Ulnar | normal | axonal | no activ | abnormal | demyel | no activ | abnormal | axonal | no activ |
| R Sural | normal | 0 | no activ | abnormal | 0 | no activ | normal | 0 | no activ |
| R Peroneal | normal | axonal | no activ | abnormal | 0 | no activ | normal | normal | no activ |
| L Median | normal | axonal | no activ | abnormal | demyel | no activ | normal | axonal | no activ |
| L Radial | normal | normal | no activ | abnormal | 0 | no activ | normal | axonal | no activ |
| L Ulnar | normal | axonal | no activ | abnormal | demyel | no activ | normal | axonal | no activ |
| L Sural | normal | axonal | no activ | abnormal | # | no activ | normal | normal | no activ |
| L Peroneal | normal | axonal | no activ | abnormal | # | 3.6/1.0 | normal | normal | no activ |
| **Patient** | | | **4** | | | **5** | | | **6** |
| **Nerve** | | | | | | | | | |
| R Median | abnormal | 0 | no activ | abnormal | 0 | no activ | abnormal | axonal | no activ |
| R Radial | abnormal | 0 | no activ | abnormal | 0 | no activ | abnormal | 0 | no activ |
| R Ulnar | abnormal | dem/ax. deg | no activ | abnormal | dem/ax. deg | no activ | abnormal | axonal | 2.9/1.1 |
| R Sural | abnormal | 0 | no activ | # | # | no activ | abnormal | 0 | no activ |
| R Peroneal | abnormal | 0 | no activ | # | # | no activ | abnormal | demyel | no activ |
| L Median | abnormal | 0 | no activ | abnormal | 0 | no activ | abnormal | demyel | no activ |
| L Radial | abnormal | 0 | 3.6/0.9 | abnormal | 0 | no activ | abnormal | 0 | no activ |
| L Ulnar | abnormal | demyel | 3.0/0.78 | abnormal | axonal | no activ | abnormal | dem/ax. deg | 2.8/1.2 |
| L Sural | abnormal | 0 | no activ | # | # | no activ | abnormal | 0 | no activ |
| L Peroneal | abnormal | 0 | no activ | # | dem/ax. deg | no activ | abnormal | 0 | no activ |
| **Patient** | | **7** | | | **8** | | | | |
| **Nerve** | | | | | | | | | |
| R Median | abnormal | 0 | no activ | normal | Normal | no activ | | | |
| R Radial | abnormal | 0 | no activ | normal | Axonal | no activ | | | |
| R Ulnar | abnormal | dem/ax. deg | no activ | normal | demyel | 2.7/0.9 | | | |
| R Sural | abnormal | 0 | no activ | normal | 0 | 2.4/1.4 | | | |
| R Peroneal | abnormal | 0 | 5.8/1.6 | normal | 0 | 2.4/0.8 | | | |
| L Median | abnormal | 0 | no activ | normal | Normal | no activ | | | |
| L Radial | abnormal | 0 | no activ | normal | Normal | no activ | | | |
| L Ulnar | abnormal | demyel | no activ | abnormal | dem/ax. deg | 3.3/0.9 | | | |
| L Sural | abnormal | 0 | 2.7/1.3 | normal | 0 | 2.4/1.4 | | | |
| L Peroneal | abnormal | 0 | no activ | normal | 0 | 1.7/0.8 | | | |

Abbreviations: #, not realized; no activ, no activity; demyel, demyelinating; dem/ax. deg., demyelinating with axonal degeneration; t-SUV max/bg-SUV max—target maximum standardized capture value/ background.

## Patient 1          Patient 3

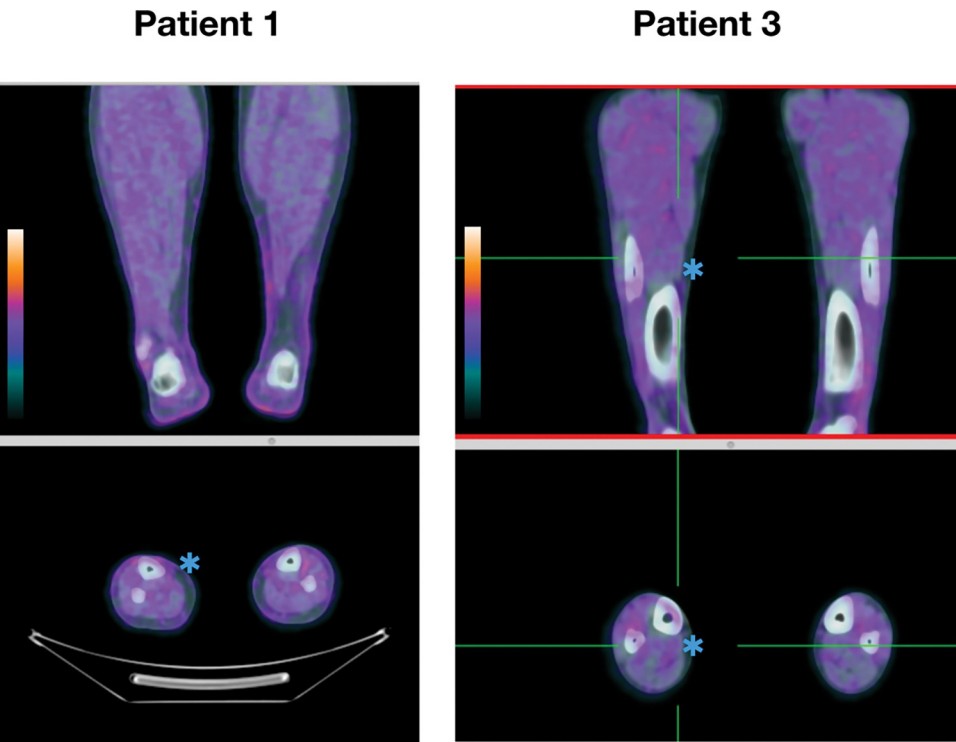

**Fig 1. Coronal and axial slices of 18F-FDG PET/CT show no uptake at the peripheral nerves (patients 1 and 3), such as the sural nerve (blue asterisk).**

by nerve conduction studies. 18F-FDG PET/CT showed hypermetabolism of radiolabeled glucose in the bilateral ulnar, sural, and superficial peroneal nerves (Fig 3).

The first six patients underwent nerve biopsy, which revealed chronic inflammatory neuropathy of leprosy etiology and positive acid-fast bacillus (AFB). Table 3 shows the relationship

## Patient 2          Patient 4          Patient 6

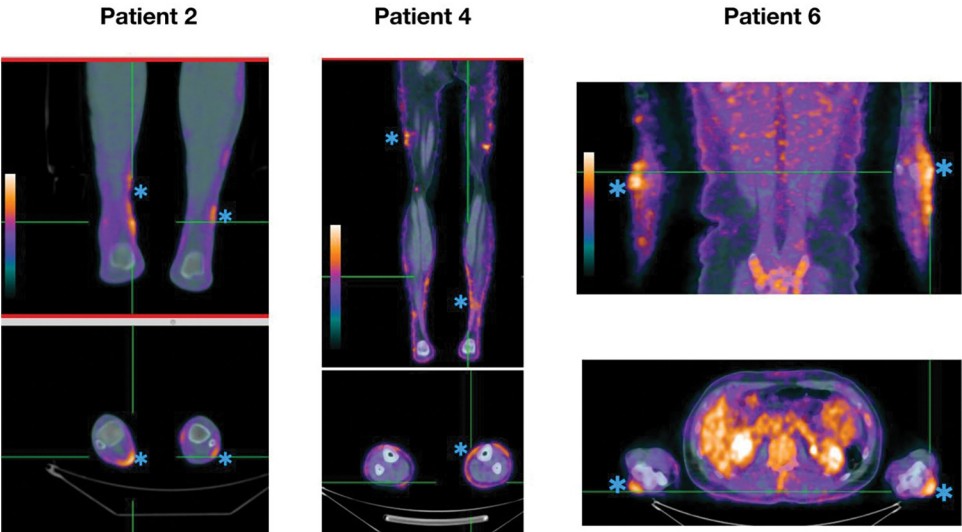

**Fig 2. Coronal and axial slices of 18F-FDG PET/CT show different levels of uptake at the peripheral nerves (blue asterisks), such as the sural and tibial nerve (patients 2 and 4), and ulnar and cutaneous nerve (patient 6), as seen at the color scale as orange to white.**

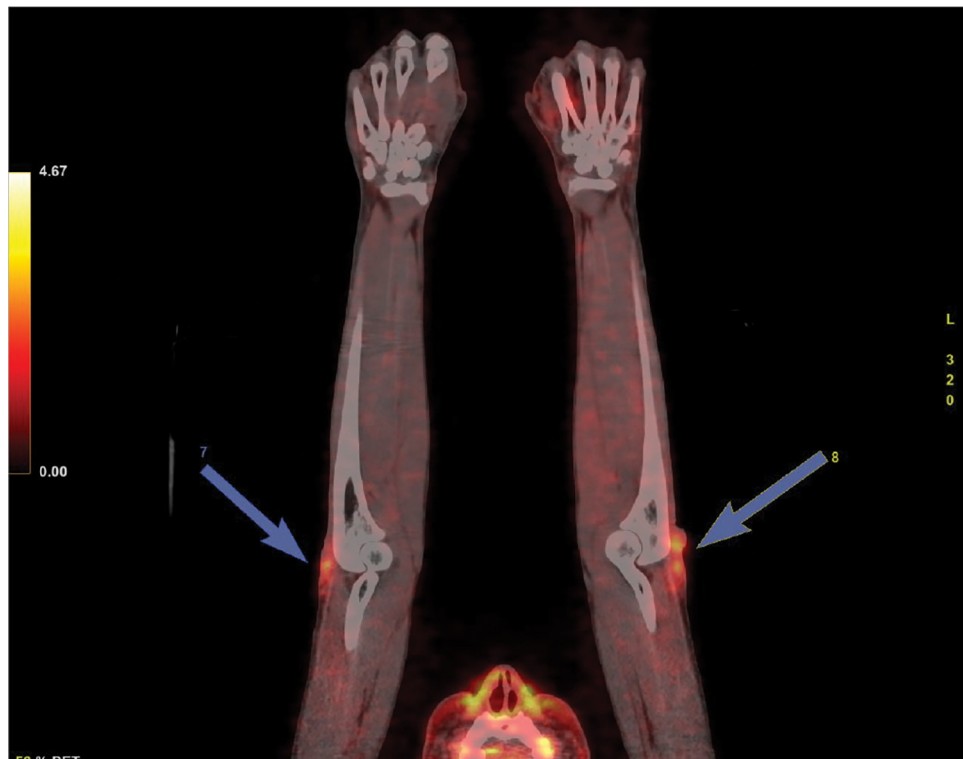

**Fig 3. Maximum intensity projection (MIP).** 18F-FDG PET/CT shows high uptake (patient 8) in bilateral ulnar (blue arrows), sural, and superficial peroneal nerves.

between 18F-FDG PET/CT uptake and the percentage of inflammatory infiltrates in the biopsied nerves.

## Discussion

Rambukkana et al. (2002) demonstrated that nerve demyelination by *M. leprae* could occur even without the inflammatory process when the bacillus promotes Schwann cell reprogramming and alters nerve function through mechanisms that need to be clarified [19]. Masaki et al. (2013) suggested that leprosy neuropathy could occur due to Schwann cell dedifferentiation after infection, caused by loss of the SOX-10 transcription factor, for example, long before leukocyte infiltration and nerve fibrosis [9].

**Table 3. Correlation between biopsy findings, NCS and PET uptake.**

| Patient | Nerve biopsied | PET uptake | NCS | %infl.inf. | fibrosis |
|---|---|---|---|---|---|
| 1 | Sural D | No | Inexcitable | 8 | Yes |
| 2 | Ulnar E | No | Inexcitable | 60 | Yes |
| 3 | Sural D | No | Inexcitable | 30 | Yes |
| 4 | Ulnar D | No | Inexcitable | 40 | Yes |
| 5 | Ulnar D | No | Inexcitable | 100 | Yes |
| 6 | Ulnar D | Yes | Inexcitable | 100 | Yes |

Abbreviations: NCS, Nerve conduction studies; %infl.inf., percentage of inflammatory infiltrate

Medeiros et al. (2016) have demonstrated increased glucose uptake in infected Schwann cells in vitro and in vivo [3]. In this research project, we chose 18F-FDG PET/CT as an imaging resource to demonstrate glucose uptake in damaged nerve tissue of LL patients with silent neuritis and acute neuritis, since *in vivo* and *in vitro* previous studies have demonstrated this metabolic alteration.

We identified that LL patients with clinically and electrophysiologically diagnosed silent neuritis had an absence or a few 18F-FDG uptakes in their nerves. Apparently, 18F-FDG PET/CT was not sensitive enough to detect metabolic changes in infected Schwann cells in cases 1, 3 and 5, nor was it sensitive enough to demonstrate the extent of nerve damage, as demonstrated in neurophysiological examinations in patients 2, 4 and 6.

Roy et al. (2018) and Shao et al. [12] reported cases in which acute leprosy neuritis was detected using 18F-FDG PET/CT [10,11]. Our findings were similar to those of patients diagnosed with acute neuritis.

It is known that, once inside the cell, the 18F-FDG molecule is phosphorylated by hexokinase, transforming it into 18F-FDG-6-Phosphate. Thus, it cannot follow the glycolytic pathway or leave the cell, which causes its uptake. Inflammatory/infectious processes with high expression of neutrophils and activated macrophages also show increased 18F-FDG uptake. Therefore, we believe that further studies are necessary regarding silent nerve injuries, where the cellular inflammatory process seems different from that of acute neuritis.

These clarifications regarding the pathophysiology of silent neuritis are essential to developing new drugs to prevent permanent disabilities in these patients.

In conclusion, 18F-FDG uptake is related to acute neuritis in LL patients, and there is no 18F-FDG uptake among LL patients who present with silent neuritis. Furthermore, 18F-FDG PET-CT could be a valuable tool to confirm neuritis. In addition, in cases of already diagnosed patients who develop recurrent neuropathy, there is a possible tool to differential diagnosis between a new episode of neuritis and chronic neuropathy.

## Acknowledgments

We thank the people who agreed to be study subjects, the team from the Leprosy Laboratory and outpatient clinic of the Instituto Oswaldo Cruz da Fiocruz, and the team from the nuclear medicine sector of Clementino Fraga Filho University Hospital. We would especially like to thank Cristiane Domingues, who was always attentive to the admitted LL patients.

## Author Contributions

**Conceptualization:** Patricia Sola Penna, Sergio Augusto Lopes De Souza, Marcia Rodrigues Jardim.

**Data curation:** Sergio Augusto Lopes De Souza, Anna Maria Sales, Roberta Olmo Pinheiro, Marcia Rodrigues Jardim.

**Formal analysis:** Patricia Sola Penna, Sergio Augusto Lopes De Souza, Roberta Olmo Pinheiro, Marcia Rodrigues Jardim.

**Investigation:** Patricia Sola Penna, Paulo Gustavo Limeira Nobre De Lacerda, Izabela Jardim Rodrigues Pitta, Clarissa Neves Spitz, Anna Maria Sales, Ana Caroline Siquara De Souza, Euzenir Nunes Sarno.

**Methodology:** Patricia Sola Penna.

**Project administration:** Marcia Rodrigues Jardim.

**Supervision:** Roberta Olmo Pinheiro, Marcia Rodrigues Jardim.

**Visualization:** Flavio Alves Lara, Euzenir Nunes Sarno.

**Writing – original draft:** Patricia Sola Penna.

**Writing – review & editing:** Patricia Sola Penna, Sergio Augusto Lopes De Souza, Flavio Alves Lara, Roberta Olmo Pinheiro, Marcia Rodrigues Jardim.

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
