## [Decision Letter · Decision Letter 0]

24 Jan 2023

Dear Dr Penna,

Thank you very much for submitting your manuscript "Evidencing Leprosy Neuronal Inflammation by 18-Fluoro-deoxy-glucose" for consideration at PLOS Neglected Tropical Diseases. As with all papers reviewed by the journal, your manuscript was reviewed by members of the editorial board and by several independent reviewers. In light of the reviews (below this email), we would like to invite the resubmission of a significantly-revised version that takes into account the reviewers' comments. 

Your manuscript has been reviewed by two experts, here and in the attachment. While they have found the manuscript interesting, they have raised some issues with the use of this technique. Please address each of their comments, both in an itemized rebuttal and in the manuscript itself. Special consideration should be given to the justification for the use of this technique in leprosy and its clinical applications.

We cannot make any decision about publication until we have seen the revised manuscript and your response to the reviewers' comments. Your revised manuscript is also likely to be sent to reviewers for further evaluation.

Sincerely,

Linda B Adams

Academic Editor

Mathieu Picardeau

Section Editor

Your manuscript has been reviewed by two experts, here and in the attachment. While they have found the manuscript interesting, they have raised some issues with the use of this technique. Please address each of their comments, both in an itemized rebuttal and in the manuscript itself. Special consideration should be given to the justification for the use of this technique in leprosy and its clinical applications.

Reviewer's Responses to Questions

**Key Review Criteria Required for Acceptance?**

**Methods**

-Are the objectives of the study clearly articulated with a clear testable hypothesis stated?

-Is the study design appropriate to address the stated objectives?

-Is the population clearly described and appropriate for the hypothesis being tested?

-Is the sample size sufficient to ensure adequate power to address the hypothesis being tested?

-Were correct statistical analysis used to support conclusions?

-Are there concerns about ethical or regulatory requirements being met?

Reviewer #1: Reject

Reviewer #2: The methods are generally satisfactory for a preliminary report. Some details need clarification: 

 The authors have made a reasonable attempt to assess histopathological findings in the nerves and to correlate these with the PET scans, but they did not (and probably could not) biopsy the same nerve sites that had been assessed by the scans. This is an inherent limitation in studying human peripheral nerves. Since leprosy neuritis and neuropathy may occur intermittently along a nerve, rather than diffusely, it is important that they acknowledge the fact that the sites are not identical and therefore that any correlations must be interpreted very cautiously. 

 “In short, an axonal lesion was defined as either an isolated reduction in amplitude ≥ 30% of the reference values”. Don’t you mean an isolated reduction in amplitude of < 30% . . .? or a 30% reduction . . .?

 Histopathological examination -- “The sensory nerve was biopsied according to clinical or electrophysiological findings” Which sensory nerve?

**Results**

-Does the analysis presented match the analysis plan?

-Are the results clearly and completely presented?

-Are the figures (Tables, Images) of sufficient quality for clarity?

Reviewer #1: (No Response)

Reviewer #2: Most of the results are clear and appropriate. some concerns:

 Describing the time of onset of nerve vs dermatological symptoms, only 6 patients are accounted for:

2 (#4 & 7) with nerve onset first

2 (5&6) with nerve and skin onset at the same time

2 (2&8) with nerve onset after skin lesions

What about cases #1 and 3?

In Table 1, ‘Symp on” refers to duration of neural symptoms. What does ‘No” mean ?

In the legend for Table 2, please also include explanations for the terms used in the headings, e.g., “t-SUV, max/bg-SUV, max”

“In patients 7 and 8, PET/CT was performed during a neuritis episode, and 18F-FDG uptake was observed in the peripheral nerves in which the patients had symptoms and signs of neuritis, as well as in other nerves that showed no signs of neuritis on neurological examination, but with neurophysiological changes.” Does the final expression, “ with neurophysiological changes”, refer to the 18F-FDg uptake? If so, it will be less confusing to say so.

Also, if I understand this correctly, this means that some of the nerves with ‘no signs of neuritis’ had increased uptake. Plese comment on this.

**Conclusions**

-Are the conclusions supported by the data presented?

-Are the limitations of analysis clearly described?

-Do the authors discuss how these data can be helpful to advance our understanding of the topic under study?

-Is public health relevance addressed?

Reviewer #1: (No Response)

Reviewer #2: The authors appear to have overstated the significance of some of their findings. Specifically, assertions regarding the assessment of Schwann cell activity/ uptake of FDG are overstated, since the methods do not appear to have sufficient sensitivity or resolution to distinguish increased uptake in Schwann cells vs inflammatory infiltrates, i.e., macrophages and lymphocytes. 

Discussion

The first paragraph is somewhat confusing and should be edited to more concisely describe the overall findings. 

e.g., “ . . . patients with LL present with sensory and motor nerve impairment despite the absence of 18F-FDG uptake in their nerves” – does this mean that they have previously had nerve injury, so they have impairment, but do not have active injury occurring at the time they present to the clinic?

“According to the present and previous studies, neuropathy in LL patients can occur as a result of Schwann cell dedifferentiation . . .” Nothing in this paper presents any evidence regarding Schwann cell dedifferentiation, and the methods do not address such dedifferentiation in any way. Therefore, this statement and reference should be omitted.

“Despite this, the nerve biopsy revealed an inflammatory process, even without clinical neuritis.” There are several possible explanations, but the most obvious is that the biopsy sites do not correspond exactly to the sites imaged by 18F-FDG uptake (see general comments). Therefore, any attempts to correlate these much be stated very cautiously. 

“Apparently, 18F-FDG PET/CT was not sensitive enough to detect metabolic changes in infected Schwann cells.“ I do not see that the authors have determined that any of the uptake seen in their own study is specifically due to Schwann cells. Is the imaging of sufficient resolution and specificity to distinguish uptake by Schwann cells vs nearby macrophages or lymphocytes?

“We believe that further studies are necessary regarding silent nerve injuries, where the cellular inflammatory process seems to be different from that of acute neuritis . . .” This assumes that an inflammatory process is present in silent nerve injury, but that has not been determined. Alternatively, it could be a degenerative process rather than an inflammatory one.

“In addition, in cases of already diagnosed patients who develop recurrent neuropathy, there is a differential diagnosis between a new episode of neuritis and chronic neuropathy” The meaning and point of this sentence is not clear.

**Editorial and Data Presentation Modifications?**

Reviewer #1: (No Response)

Reviewer #2: Author summary

Typos and incomplete sentences in the following: “ . . . our question was about how the uptake of this radiopharmaceutical would be in a common silent neuritis in LL patients.d the uptake . . .” 

The phrase real extension seems odd and I don’t understand what this means here.: ´. . . Our data demonstrated, for the first time, the real extension of peripheral nerve . . .’ I think you mean something else -- Perhaps you mean illustration of . . . or example of 

Introduction 

“Medeiros et.al. (2016) evaluated in vitro and in vivo models of Schwann cells . . .” This should be indicated as reference #3 instead of the year. 

The expression “real extension” also appears in the last paragraph of the introduction.

**Summary and General Comments**

Reviewer #1: (No Response)

Reviewer #2: This is an interesting report regarding the use of 18FDG-PET as a means to assess physiological activity, in a small series of patients. The addition of this method to other tools for neurological assessment enables measurement of physiological activity in real time, a valuable new approach. 

The peripheral nerve status of the patients has been carefully assessed and well documented using a variety of clinical assessments in addition to the PET scans. The finding of increased uptake during active neuritis is not surprising, but the capability to assess physiological activity in vivo during acute neuritis offers new possibilities in understanding this process. The finding of no uptake in patients with neuropathy but not acute neuritis also adds to our knowledge of neuropathy in leprosy. 

The findings in acute neuritis do increase the evidence base regarding the inflammation in nerves in leprosy. It may be valuable to consider this in the broader context of the findings of FDG uptake in other inflammatory conditions such as arthritis, e.g., Graham RN, Panagiotidis E. [18F]FDG PET/CT in rheumatoid arthritis. Q J Nucl Med Mol Imaging. 2022 Sep;66(3):234-244.

It would be interesting to know if the authors believe that 18FDG-PET could be a useful tool to measure changes in inflammation during treatment of acute neuritis in leprosy, i.e., scanning before and during prednisone treatment.

PLOS authors have the option to publish the peer review history of their article (what does this mean?). If published, this will include your full peer review and any attached files.

Reviewer #1: No

Reviewer #2: Yes: David Scollard, M.D., Ph.D.
---

## [Editor Report · Decision Letter 1]

20 Apr 2023

Dear Dr Penna,

Thank you very much for submitting your manuscript "Evidencing Leprosy Neuronal Inflammation by 18-Fluoro-deoxy-glucose" for consideration at PLOS Neglected Tropical Diseases. As with all papers reviewed by the journal, your manuscript was reviewed by members of the editorial board and by several independent reviewers. In light of the reviews (below this email), we would like to invite the resubmission of a significantly-revised version that takes into account the reviewers' comments. 

The original manuscript was sent to two highly qualified and knowledgeable reviewers. However, some of their comments were not addressed at all in the rebuttal; it is imperative to provide a response to every comment. In addition, each of their concerns must be addressed in the manuscript itself. In the rebuttal, please include the line numbers of the manuscript where each of the specific concerns were edited and clarified. Modifications to address their comments will improve the manuscript and reduce misunderstandings by other readers.

We cannot make any decision about publication until we have seen the revised manuscript and your response to the reviewers' comments. Your revised manuscript is also likely to be sent to reviewers for further evaluation.

Sincerely,

Linda B Adams

Academic Editor

Mathieu Picardeau

Section Editor

This manuscript was sent to two highly qualified and knowledgeable reviewers. However, some of their comments were not addressed at all in the rebuttal; it is imperative to provide a response to every comment. In addition, each of their concerns must be addressed in the manuscript itself. In the rebuttal, please include the line numbers of the manuscript where each of the specific concerns were edited and clarified. Modifications to address their comments will improve the manuscript and reduce misunderstandings by other readers.
---

## [Editor Report · Decision Letter 2]

16 May 2023

Dear Dr Penna,

We are pleased to inform you that your manuscript 'Evidencing Leprosy Neuronal Inflammation by 18-Fluoro-deoxy-glucose' has been provisionally accepted for publication in PLOS Neglected Tropical Diseases.

Best regards,

Linda B Adams

Academic Editor

Mathieu Picardeau

Section Editor

---

## [Editor Report · Acceptance letter]

1 Jun 2023

Dear Dr Penna,

We are delighted to inform you that your manuscript, "Evidencing Leprosy Neuronal Inflammation by 18-Fluoro-deoxy-glucose," has been formally accepted for publication in PLOS Neglected Tropical Diseases.

Best regards,

Shaden Kamhawi

co-Editor-in-Chief

Paul Brindley

co-Editor-in-Chief
